# Identification of an Autophagy-Related Signature for Prognosis and Immunotherapy Response Prediction in Ovarian Cancer

**DOI:** 10.3390/biom13020339

**Published:** 2023-02-09

**Authors:** Jinye Ding, Chunyan Wang, Yaoqi Sun, Jing Guo, Shupeng Liu, Zhongping Cheng

**Affiliations:** 1Department of Obstetrics and Gynecology, Shanghai Tenth People’s Hospital, Tongji University School of Medicine, Shanghai 200072, China; 2Institute of Gynecological Minimally Invasive Medicine, Tongji University School of Medicine, Shanghai 200072, China

**Keywords:** autophagy, immune microenvironment, immunotherapy, prognosis, ovarian cancer

## Abstract

Background: Ovarian cancer (OC) is one of the most malignant tumors in the female reproductive system, with a poor prognosis. Various responses to treatments including chemotherapy and immunotherapy are observed among patients due to their individual characteristics. Applicable prognostic markers could make it easier to refine risk stratification for OC patients. Autophagy is closely implicated in the occurrence and development of tumors, including OC. Whether autophagy -related genes can be used as prognostic markers for OC patients remains unclear. Methods: The gene transcriptome data of 374 OC patients were downloaded from The Cancer Genome Atlas (TCGA) database. The correlation between the autophagy levels and outcomes of OC patients was identified through the single sample gene set enrichment analysis (ssGSEA). Recognized molecular markers of autophagy in different clinical specimens were detected by immunohistochemistry (IHC) assay. The gene set enrichment analysis (GSEA), ESTIMATE, and CIBERSORT analysis were applied to explore the correlation of autophagy with the tumor immune microenvironment (TIME). Single-cell RNA-sequencing (scRNA-seq) data from seven OC patients were included for characterizing cell-cell interaction patterns of autophagy-high or low tumor cells. Machine learning, Stepwise Cox regression and LASSO-Cox analysis were used to screen autophagy hub genes, which were used to establish an autophagy-related signature for prognosis evaluation. Four tumor immunotherapy cohorts were obtained from the GEO (Gene Expression Omnibus) database and the literature for autophagy risk score validation. Results: The autophagy levels were closely related to the prognosis of the OC patients. Additionally, the autophagy levels were correlated with TIME status including immune score, and immune-cell infiltration. The scRNA-seq analysis found that tumor cells with high or low autophagy levels had different interactions with immune cells, especially macrophages. Eight autophagy-hub genes (ZFYVE1, AMBRA1, LAMP2, TRAF6, PDPK1, ATG2B, DAPK1 and TP53INP2) were screened for an autophagy-related signature. According to this signature, higher risk score was correlated with poor prognosis and better immunotherapy response in the OC patients. Conclusions: The autophagy-related signature is applicable to predict the prognosis and immune checkpoint inhibitors (ICIs) therapy efficiency in OC patients. It is possible to identify OC patients who will respond to ICIs therapy and have a favorable prognosis, although more verification is needed.

## 1. Introduction

Lacking robust screening methods and effective treatment, OC remains one of the most lethal gynecologic malignant tumors. More than 300,000 new cases of ovarian cancer were diagnosed and 190,000 deaths were expected in 2020 worldwide [1]. In total, 70–80% of patients with advanced ovarian cancer develop chemotherapy-resistant disease after traditional therapy, including cytoreductive surgery and platinum-based chemotherapy, conferring a 40% five year survival rate [2]. Novel therapeutic strategies, such as the poly-ADP ribose polymerase inhibitors (PARPis) and ICIs improved the prognosis for a subset of OC patients, but the response rate remains limited [3,4]. Currently, there is a significant need for prognostic markers that stratify OC patients’ prognoses to facilitate individualized treatment.

Autophagy is an evolutionarily conserved catabolic intracellular process in which cytoplasmic macromolecules, aggregated proteins, damaged organelles or pathogens are delivered to lysosomes for degradation [5,6]. It plays an important role in cellular development, function and homeostasis [7]. Disturbances in autophagy lead to neurodegenerative diseases, metabolic diseases and cancer [8]. Previous evidence demonstrated that autophagy levels were correlated with prognosis in tumor patients. Hu et al. reported that autophagy-related protein five (ATG5) was negatively associated with survival outcomes in colorectal cancer (CRC) patients [9]. Yang et al. found that high levels of the key autophagy protein LC3 indicated poor prognosis in pancreatic cancer patients [10]. Further studies also reported the negative role of autophagy in the prognoses of different tumor types [11,12,13]. However, the function of autophagy-related genes in the context of OC prognosis is poorly understood.

Autophagy suppresses or promotes tumor development, depending on several factors, including nutrient availability, immune activation or immune suppression in the tumor microenvironment (TME) [5,14,15,16,17]. Compared with normal cells, tumor cells are more reliant on autophagy to adapt their rapid growth rates, altered metabolism, and nutrient-deprived growth environment [18]. Emerging evidence has shown that autophagy contributes to anti-tumor immune responses and affects tumor immunotherapy. Elevated autophagy levels facilitate tumorigenesis and immune escape in non-small-cell lung cancer (NSCLC) [19]. Furthermore, autophagy inhibition increases the levels of surface MHC-Ⅰ, leading to improved antigen presentation and enhanced anti-tumor T cell response in pancreatic ductal adenocarcinoma (PDAC) [20]. It was also shown that the combination of autophagy inhibition and dual immune-checkpoint blockade increased the anti-tumor immune response [20]. Nevertheless, the exact role of autophagy in the immune microenvironment of OC, and whether autophagy-related genes can predict ICIs therapy efficiency in OC, remains unknown.

In the present study, we describe the close association between autophagy levels and the prognosis and immune microenvironment of OC via the bioinformatic analysis and IHC validation. The interaction between tumor cells and immune cells, particularly macrophages identified by scRNA-seq, was associated with the autophagy levels of tumor cells, which may help to further clarify the correlation between autophagy levels and patient prognosis. Next, an autophagy-related signature composed of eight autophagy-hub genes was established to evaluate the prognoses of OC patients. Higher scores of the signature were correlated with poor prognosis and better responses to ICIs therapy in OC, suggesting its potential availability in clinic for the evaluation of prognosis and ICIs therapy responses. In general, our research provides a novel approach to predict prognosis and ICIs therapy efficiency for OC patients based on an autophagy-related signature.

## 2. Materials and Methods

### 2.1. Acquirement of Target Datasets

The RNA-seq profiles (fragments per kilobase million (FPKM) value) and related clinical information of 374 OC patients were obtained from The Cancer Genome Atlas (TCGA) data portal through the University of California Santa Cruz (UCSC, https://xenabrowser.net/, accessed on 1 June 2021). From the Gene-Expression Omnibus (GEO) databases, two OC datasets (GSE14407, GSE38666) were enrolled for validation cohorts. Patients without complete survival information were excluded. The scRNA-seq data from seven OC patients with informed consent was from our department. Immunotherapy cohorts including IMvigor210 and an institutional cohort were downloaded according to a previous report [21,22]. The other two immunotherapy cohorts (GSE78220, GSE176307) were acquired from GEO database.

### 2.2. Gene Set Variation Analysis (GSVA)

The Hallmark gene sets of autophagy were acquired from Molecular Signatures Database (MsigDB, https://www.gsea-msigdb.org/gsea/msigdb/, accessed on 1 June 2021). Next, the single-sample gene set enrichment analysis (ssGSEA) algorithm (‘GSVA’ R package) was applied to estimate the abundance of gene signatures for autophagy. An expression matrix of autophagy-related genes was used to perform principal component analysis (PCA) on OC patients.

### 2.3. Human Clinical Specimens

The clinical tissue samples of OC were collected in two parts. The first part was a total of 6 common tumor specimens from our hospital. Normal ovary samples were included as controls. The second part was a total of 6 chemo-sensitive samples and 6 chemo-resistant samples. All samples were collected with the patients’ informed consent. This study was approved by the Ethics Committee of Shanghai Tenth People’s Hospital Health Authority (SHSY-ICE-5.0/22K76/P01).

### 2.4. Immunohistochemical Staining and Analysis

Immunohistochemistry was conducted following the method in a previous report [23]. Antibodies including LC3B (1:600, #ab192890, Abcam, UK), p62 (1:100, #ab207305, Abcam, UK), Beclin1 (1:700, #ab207612, Abcam, UK), CD86 (1:400, #ab220188, Abcam, UK), iNOS (1:100, #ab115819, Abcam, UK), and CD163 (1:1000, #16646-1-AP, Pro-teintech, China) were used to detect protein expression in tissues. Images were captured at 100x and 400x magnification with the same parameters, and five fields of view per sample were randomly selected to assess the expression levels of the protein. Next, the percentage and the staining intensity of positive cells in each image were analyzed automatically by the “IHC profiler” plugin in Image J software (Version: 1.52a). H-score (H-score = (1 × (% of cells 1+) + 2 × (% of cells 2+) + 3 × (% of cells 3+)), 1 = lack or weak expression, 2 = moderate expression, 3 = strong expression) was applied to quantify the IHC images [24,25,26]. The average H-score from five fields of view was defined as the final score of the sample. Samples with poor immunohistochemical staining were removed.

### 2.5. Analysis of Functional Enrichment and Immune Cell Infiltration in TIME

To determine the variances of immune-related signaling between autophagy high and low groups, Kyoto Encyclopaedia of Genes and Genomes (KEGG) pathway enrichment and gene set enrichment analysis (GSEA) were performed through the R package ClusterProfiler. The pathways were selected according to a rigorous q-value cutoff of 0.05. The immune score of each sample was calculated by the ‘ESTIMATE’ R package. Further, the percentage of infiltrating immune cells was evaluated by CIBERSORT in autophagy high and low subgroups.

### 2.6. Cluster Identification and Annotation of scRNA-Seq Data

Gene expression matrix of high quality was analyzed with the R package Seurat to perform cell normalization and cell filtering according to the mitochondrial metrics percentage, minimum and maximum gene numbers. Cells were removed with detected genes less than 200. Top 2000 highly variable genes were selected for further clustering analysis. A PCA was performed on the resulting matrices and top 20 PCAs were further used to reduce dimensionality using uniform manifold approximation and projection (UMAP). Clusters were annotated based on their marker genes and canonical cell markers from CellMarker website (http://biocc.hrbmu.edu.cn/CellMarker/index.jsp, accessed on 1 June 2021). The autophagy scores of malignant cells were obtained in AddModulescore function in Seurat. Next, clusters ranked top three and bottom three autophagy scores were extracted and grouped as autophagy-high and low tumor cells. Other analyses were also conducted with corresponding functions from Seurat suite.

### 2.7. Cell-Cell Interaction Analysis

Cell–cell communications at the molecular level between tumor cells and macrophages were investigated using CellPhoneDB [27]. Curated ligand-receptor interaction pairs from the built-in database of the package were utilized. Interaction pairs with *p*-values < 0.05 returned by CellPhoneDB were selected for the evaluation of cross-talk between cell subgroups.

### 2.8. Construction and Validation of Autophagy-Risk Signature

The random forest (RF) method was applied to screen out autophagy hub genes. Relative importance of these hub genes was evaluated by Gini index. Next, stepwise multivariate cox regression analysis was performed to further filter the independent variables; eight autophagy-risk genes were obtained. Based on these eight genes, LASSO analysis was conducted to construct an autophagy-risk model to evaluate the significance of autophagy-risk signature in the prognosis of OC patients. Regression coefficients and the expression levels of eight risk genes were used to compute the risk scores for each sample. The specific calculation formula is 0.2867×ZFYVE1 expression + 0.2394×AMBRA1 expression +0.1966×LAMP2 expression + (−0.4344)×TRAF6 expression + 0.2297×PDPK1 expression + (−0.4067)×ATG2B expression + 0.1523×DAPK1 expression + (−0.1447)×TP53INP2 expression.

### 2.9. Prognostic Value and Clinical Applicability Assessment of Autophagy-Risk Signature

Kaplan-Meier survival analysis was carried out to evaluate the survival outcomes of OC patients in the autophagy high-risk and low-risk groups. The accuracy of this predictive model was assessed by the time-dependent receiver operating characteristic (ROC) curve through the R package ‘time ROC’. In addition, using the R package ‘survival’, univariate and multivariate regression analyses were conducted to verify whether ‘risk score’ could be an independent prognostic factor for patients with OC. Subsequently, based on the available prognosis-related clinical features, a nomogram was plotted via the R package ‘rms’. Calibration charts at 3, 5, and 10 years were generated to assess the predictive capability of nomogram.

### 2.10. Statistical Analysis

All data were processed in the R programming environment (version 4.0). Single comparison between two groups was analyzed through a two-tailed Student’s t-test or Wilcoxon rank-sum test. Spearman’s correlations were used to determine the significance and correlation coefficients between two continuous variables. *p* < 0.05 was considered statistically significant.

## 3. Results

### 3.1. Autophagy Levels Were Highly Relevant to the Prognosis of OC Patients

Autophagy is a form of programmed cell death (PCD) and is closely associated with tumor development [28]. To explore the role of autophagy in the progression of OC, we self-clustered OC patients according to the gene expression profile matrix, and found that the patients could be divided into two clusters with significantly different levels of autophagy through PCA. However, no such result was observed in three other types of PCD, including apoptosis, necroptosis and ferroptosis (Figure 1A). The result above indicated that autophagy played an important role in OC, and patients with different autophagy levels might have diverse biological characteristics. Further, we found that patients with higher levels of autophagy had a poorer prognosis by Kaplan-Meier survival analysis (*p* = 0.0079, Figure 1B). The survival time of the OC patients showed no difference between high and low levels of apoptosis, necroptosis and ferroptosis (Appendix A). Moreover, a higher proportion of lymphatic metastases occurred in patients at high levels of autophagy compared to those with low levels of autophagy (Figure 1C). To verify the autophagy levels in the tumor tissues, immunohistochemical assays were performed for autophagy markers (LC3B, p62 and Beclin1). The results showed that the expression levels of LC3B and Beclin1 were elevated in the OC tissues compared to the normal ovarian tissues, while the expression levels of p62 were decreased (Figure 1D). Additionally, cisplatin-sensitive and cisplatin-resistant OC tissues were also included for immunohistochemical assays due to the well-recognized differences between them in the prognosis of OC patients. It was shown that the expression levels of LC3B and Beclin1 were higher in the cisplatin-resistant OC tissues than in the cisplatin-sensitive OC tissues and the expression levels of p62 were reversed (Figure 1E). Taken together, these results revealed that autophagy levels were significantly and negatively correlated with prognosis in OC patients.

### 3.2. Autophagy Was Significantly Correlated with the Immune Microenvironment in OC

It has been shown that autophagy plays an important role in the immune microenvironment of tumors [29]. Hence, we hypothesized that autophagy affected the prognoses of patients by influencing the immune microenvironment in OC. To test this assumption, we performed the GSEA of autophagy-related genes and found that the immune-related signals were positively enriched in the autophagy-high group (Figure 2A). We then performed differential expression gene identification between autophagy-high and low groups. The expression levels of immune-related genes (CD28, CD4, CD274, CTLA4, TLR2, TLR4, TICAM2, LCP2, ITK, PTPRC, EGF, JAK1 and mTOR) were upregulated in the autophagy-high group (Figure 2B,C). Other immune-related genes were also significantly higher in the autophagy-high subgroup (Figure 2C and Appendix A). These results preliminarily suggested that autophagy had an important effect on immunity in OC. Further, the ESTIMATE analysis was conducted, which demonstrated that the immune score was elevated in the autophagy-high subgroup compared to that in the autophagy-low group. (*p* = 0.024, Figure 3A). The CIBERSORT analysis was applied to evaluate the infiltration proportion of twenty-two immune cell types in the OC cohorts from the TCGA. The results illustrated that the levels of infiltrating naïve B cells, CD4 memory resting T cells, and M2 macrophages were significantly higher in the autophagy-high group (Figure 3B). The infiltration levels of memory B cells, CD8 T cells, and gamma-delta T cells were lower in the autophagy low group. The infiltration difference was further verified by the correlation analysis (Figure 3C). Moreover, the M1/M2 macrophages that infiltrated into the tissues were investigated via IHC due to their most significant difference in the CIBERSORT analysis. Signals of iNOS (M1 macrophage) showed no difference between the tumors and the normal tissues, while more and enhanced CD163 signals (M2 macrophages) were observed in the tumors (Figure 3D) [30]. Higher CD163 expression was also found in the cisplatin-resistant OC tissues compared with the cisplatin-sensitive tissues when the CD86 (M1 macrophage) expression was similar among the tissues (Figure 3E) [31,32]. These data suggested a correlation between autophagy level and OC immune microenvironment.

### 3.3. Macrophages Contributed to the Autophagy-Involved Immune Microenvironment of OC

Having found significant enrichment of macrophages in autophagy-high tissues, we intended to investigate the interaction between macrophages and tumor cells with different autophagy levels. Firstly, each tumor cell from seven tumor tissues was assigned an autophagy score based on the gene expression from the scRNA-seq. The UMAP plots showed that the clusters had different autophagy scores (Figure 4A,B), suggesting these cells were at different levels of autophagy. According to the autophagy score, tumor cells were identified as high or low autophagy levels. Furthermore, the interactions between the tumor cells and the macrophages were investigated via ligand-receptor interaction evaluation. More ligand-receptor interactions were observed between the autophagy-high tumor cells and the macrophages (Figure 4C). Receptor-ligand pairs, including CD74_APP, CD74_COPA, MDK_LRP1, and TNFRSF1A_GRN, were regarded as displaying strong interaction (Figure 4C). Functional macrophages are reported to contain two polarization states: alternatively activated M2 and classically activated M1-like subtypes. The former exerts a tumor-promoting role involving immunosuppression, while the latter has an anti-tumor effect due to its intrinsic phagocytic properties [33]. There were two macrophage clusters, which were identified to be M1-like and M2-like based on gene expression (Figure 4D,E). Accordingly, the cell-cell interaction analysis found different receptor-ligand pairs among tumor cells. The interaction pairs via SPP1, RARRES2, NECTIN3, LGALS9, and LAMB1 showed increased signaling in the autophagy-high tumor cells, while SEMA3C, PTN, ICAM1, GAS6, and CSF1 showed decreased signaling compared with those in the autophagy-low tumor cells (Figure 4F). This suggested a correlation between tumor-cell autophagy level and macrophage polarization, since the signals mentioned above contributed to macrophage polarization [34,35,36,37,38,39,40,41]. These data indicated that autophagy might affect the immune microenvironment of OC by encouraging macrophage polarization.

### 3.4. Establishment of Autophagy-Related Signature

To translate our evidence into a tool to support clinical practice, we attempted to construct an autophagy-related signature based on autophagy-hub genes. Firstly, thirty hub genes associated with autophagy were screened out by random forest (RF) (Appendix A). These genes were ranked in accordance with their importance in the autophagy process (Figure 5A). Then, to make the signature more streamlined, stepwise multivariate cox proportional risk regression analysis was performed and used to filter out eight risk genes (ZFYVE1, AMBRA1, LAMP2, TRAF6, PDPK1, ATG2B, DAPK1, and TP53INP2) (Figure 5B). To evaluate the significance of these eight risk genes in the signature for the prognosis of OC patients, the risk scores for each case were calculated based on the expression levels and regression coefficients of the risk genes. Subsequently, the OC patients in the TCGA cohort were divided into high-risk (*n* = 186) and low-risk (*n* = 186) groups according to the median risk score. The Kaplan-Meier survival analysis showed that the overall survival (OS) of the patients (TCGA) in the high-risk group was significantly shorter than that of in the low-risk group (*p* < 0.05) (Figure 5C). The time-dependent ROC curves analysis showed that the prognostic accuracies of the signature were 0.602 at 3 years, 0.628 at 5 years and 0.808 at 10 years (Figure 5D). We also verified the prognostic value of the autophagy-related signature in two other GEO cohorts and obtained the same results as shown in Figure 5C (Figure 5E). Moreover, the autophagy-risk scores of the OC patients in the TCGA cohort were ranked, and we analyzed their distribution (Figure 5F). The patients’ survival status and the expression levels of the eight risk genes in the high-risk and low-risk groups were described in Figure 5G, H, respectively.

### 3.5. Autophagy-Related Signature Was an Independent Prognostic Predictor

Next, we explored whether the autophagy-related signature could predict the prognosis of OC patients independently of other risk factors. A univariate cox regression analysis showed that, except for the FIGO stage (III-IV), both age and autophagy-risk score were closely associated with the prognoses of OC patients in the TCGA cohort (*p* < 0.05, Figure 6A). A multivariate cox regression analysis revealed that both age and autophagy-risk score were independently correlated with OS in the patients with OC in the TCGA cohort (*p* < 0.05, Figure 6B). These results showed that autophagy-related signature was an independent predictor of prognoses in the OC patients. To visualize the relationship between the clinical prognostic factors and the survival probability, and thus guide clinicians towards better prognostic predictions, a nomogram was constructed. We observed that the patients with higher total points had worse survival outcomes (Figure 6C). The C-index of the nomogram was 0.623 (95% confidence interval, 0.584–0.662). The calibration curves showed that the OS predicted by the nomogram at 3, 5, and 10 years was in good accordance with the OS actually observed (Figure 6D).

### 3.6. Autophagy-Related Signature Predicted ICIs Therapy Efficiency of OC

Since autophagy was closely correlated with the OC immune microenvironment, we next evaluated whether autophagy-related signature was associated with the response to ICIs therapy. The clinical trial IMvigor210, which explored the therapeutic effects of the anti-PD-L1 immunotherapy in urothelial cancer (UC) that had similar histological type to OC [42], was used for the following study. The patients who responded to anti-PD-L1 immunotherapy in the IMvigor210 cohort had a higher autophagy-risk score based on our signature (Figure 7A). This result was validated in an anti-PD-1 immunotherapy cohort (renal cell carcinoma) [22]. The results showed that the patients with partial response (PR) to anti-PD-1 treatment displayed higher autophagy-risk scores than patients with progressive disease (PD) (Figure 7B). However, the autophagy-risk scores did not show significant differences among the patients with a complete response (CR), PD, PR and stable disease (SD) in other immunotherapy cohorts (GSE176307_(UC) and GSE78220_(melanomas)) (Figure 7C,D). Previous studies have shown that tumor mutational burden (TMB) and tumor neoantigen burden (TNB) reflected the ability of tumor cells to produce neoantigen, and that the higher the TMB and TNB values, the greater the benefit for the patient from ICIs therapy. The combination of the two markers was used as a predictor of ICIs therapy efficacy for a wide range of tumors [43,44]. Therefore, we assessed the correlation between the autophagy risk score and these two well recognized indexes. The results demonstrated that the group with the higher autophagy-risk scores had higher TMB and TNB values in the IMvigor210 cohort. This finding indicated that patients with higher autophagy-risk scores may have better responses to anti-PD-L1 immunotherapy. In a word, the results above suggested that the autophagy-risk score based on the autophagy-related signature could assist in predicting the response rates of patients to ICIs therapy.

## 4. Discussion

OC is one of the most malignant tumors of the female reproductive system, with late presentation and high recurrence, remaining the leading cause of mortality among all gynecological malignancies. Usually, OC is treated with surgery and chemotherapy, but with poor outcomes. ICIs therapy has brought a breakthrough in the field of solid tumor immunotherapy. However, the response rates among OC patients remain modest due to its complex TIME. Hence, the discovery of new biomarkers and potential therapeutic targets is essential to improve the prognoses of patients with OC. In the present study, we firstly explored and validated the correlation of autophagy with prognosis and immune cell infiltration in OC. Then, the effect of high and low levels of autophagy in tumor cells on macrophage polarization was further analyzed using the scRNA-seq analysis. Considering the impact of autophagy levels on the prognosis and the immune microenvironment of OC, we constructed an autophagy-related signature and assessed its prognostic value in OC cohorts. Finally, we evaluated the utility of this signature in predicting the response rate to ICIs therapy. Taken together, our results demonstrated that the autophagy-related signature was a promising supportive tool in the prediction of prognoses and ICIs therapy efficiency in OC patients, providing a new direction for the individualized treatment of OC.

In this study, the negative correlation between autophagy levels and the prognoses of OC patients was determined according to the results of bioinformatic analyses and IHC validation. It was worth mentioning that significant differences in autophagy levels were observed not only in normal ovarian and OC tissues, but also in cisplatin-sensitive and cisplatin-resistant OC tissues. This result which indicated the elevated autophagy levels in cisplatin-resistant OC tissues was consistent with those of other studies [6,7,45,46]. On this basis, we hypothesized that the nucleolar stress in cancer cells was one of the important contributors. Previous evidence demonstrated that cisplatin killed cells by inhibiting ribosome biogenesis, leading to nucleolar stress [47]. Nucleolar stress is usually mediated by diverse ribosome proteins and/or nucleolar proteins, such as RPLP0, RPLP1, RPLP2, and uL3 [48]. Pecoraro et al. showed that the depletion of uL3 increased the drug resistance of colon tumor cells by activating autophagy [49]. Therefore, it was reasonable to suppose that elevated levels of autophagy in cisplatin-resistant OC tissues might be involved in the above factors.

This study identified eight autophagy-risk genes and constructed an autophagy-related signature based on these eight genes. We then validated the prognostic value of the signature in the TCGA cohorts of OC. Reviewing the eight genes, ZFYVE1 is a guanylate-binding protein which contains a zinc-finger FYVE domain. It is usually identified as a molecular marker of omegasome [50]. In recent years, ZFYVE1 has been found to play an important role in the innate immune and inflammatory response [51]. Autophagy regulator AMBRA1 is involved in a variety of biological processes, including autophagy, proliferation and apoptosis, and it exhibits anti-tumor effects in melanoma [52]. LAMP2 is an extremely important lysosomal membrane protein, which accounts for approximately 50% of all the proteins in the lysosome membrane [53]. It is also a well-established mediator of autolysosome maturation [54]. TRAF6 has ubiquitin ligase activity and it regulates TLR4-induced autophagy by ubiquitinating BECN1 [55]. It was reported that the master kinase PDPK1 bound to AKT and suppressed autophagy by activating AKT-mTOR signaling [56]. ATG2B is a lipid transporter protein that is required in the formation of autophagosomes and in the regulation of lipid droplet morphology [57]. DAPK1 belongs to a family of serine/threonine protein kinases and functions as a tumor suppressor. Furthermore, it is a key regulator of autophagy [58]. TP53INP2 is initially localized in the nucleus, and upon translocation to the cytoplasm it can bind to autophagosome and subsequently promote autophagy [59]. These studies have demonstrated the critical role of the eight genes during the process of autophagy; nevertheless, their specific biological role in OC needs to be investigated further.

The complex TME of OC is a major influencer of the prognoses of OC patients. Besides tumor cells, the TME also contains tumor-associated fibroblasts, endothelial cells and infiltrating immune cells [60]. The levels of immune cell infiltration in solid tumors are extremely important in tumor development. In this study, the effect of autophagy on the infiltration levels of different immune cells was analyzed. We observed a significantly higher immune score in the autophagy-high group compared to the autophagy-low group. The infiltration ratio of naive B cells, memory resting CD4+ T cells and M2 macrophages was greater in the autophagy-high group. Conversely, memory B cells, CD8+ T cells, follicular helper T cells and gamma delta T cells were more abundant in the autophagy low group. In particular, the proportion of infiltrating M2 macrophages varied dramatically at different levels of autophagy. It has been reported that autophagy can cause M2-like macrophage polarization in chronic inflammatory diseases [61]. However, the impact of autophagy on macrophage polarization in OC is unknown. We explored the interaction pattern between tumor cells with different autophagy levels and macrophages through the scRNA-seq data. The OC cells with high autophagy levels showed an increased interaction with macrophages through SPP1-CD44, RARRES2-CMKLR1, NECTIN3-NECTIN2, LGALS9-CD44, and LAMB1-CD44 signaling, and decreased interaction via SEMA3C-PLXND1, PTN-SDC4, ICAM1-(ITGAX+ITGB2), GAS6-MERTK and CSF1-CSF1R signaling. Emerging evidence demonstrated that the ligand-receptor signaling described above was closely correlated with macrophage polarization [34,35,36,37,38,39,40,41]. For example, Zhang et al. reported that A549 cells were able to promote the M2 polarization of THP-1 macrophages though the induction of SPP1 [34]. In summary, these findings indicated that autophagy might play a key role in the immune microenvironment of OC by affecting macrophage polarization.

It has been reported that autophagy levels can affect the efficiency of ICIs therapy [62]. We found that the patients in IMvigor210 who responded to immunotherapy had significantly higher autophagy-risk scores compared to those who had no response. Consistent results were found in another anti-PD-1 immunotherapy cohort regarding RCC. Additionally, higher TMB and TNB values were observed in the patients with a high risk of autophagy in the IMvigor210 cohort. This was in accordance with the results that autophagy was involved in the T-cell meditated immune response in tumors with high TMB, but not in tumors with low TMB [63]. However, no significant differences in autophagy- risk score were observed among the patients with different responses in two other immunotherapy cohorts (GSE176307 and GSE78220). We speculated that this might be caused by the insufficient sample size or the specificity of tumor tissues. Nevertheless, all the findings suggested that patients at high risk of autophagy were more likely to benefit from ICIs therapy. On this basis, we also considered the factors that potentially contributed to this result. As is well known, the heterogeneity of cancer cells is an important factor in the pathology of cancer and therapy response [64]. In this study, the high/low autophagy levels in the tumor cells allowed them to present different phenotypes, and thus they displayed different responses to immunotherapy. This reminds us that it is indispensable to explore the heterogeneity of cancer cells in greater depth.

## 5. Conclusions

In summary, our study successfully identified a novel autophagy-related signature, which not only helps to evaluate the prognoses of OC patients, but also potentially aids in distinguishing the responses to ICIs therapy of patients with specific cancer types.

## Figures and Tables

**Figure 1 biomolecules-13-00339-f001:**
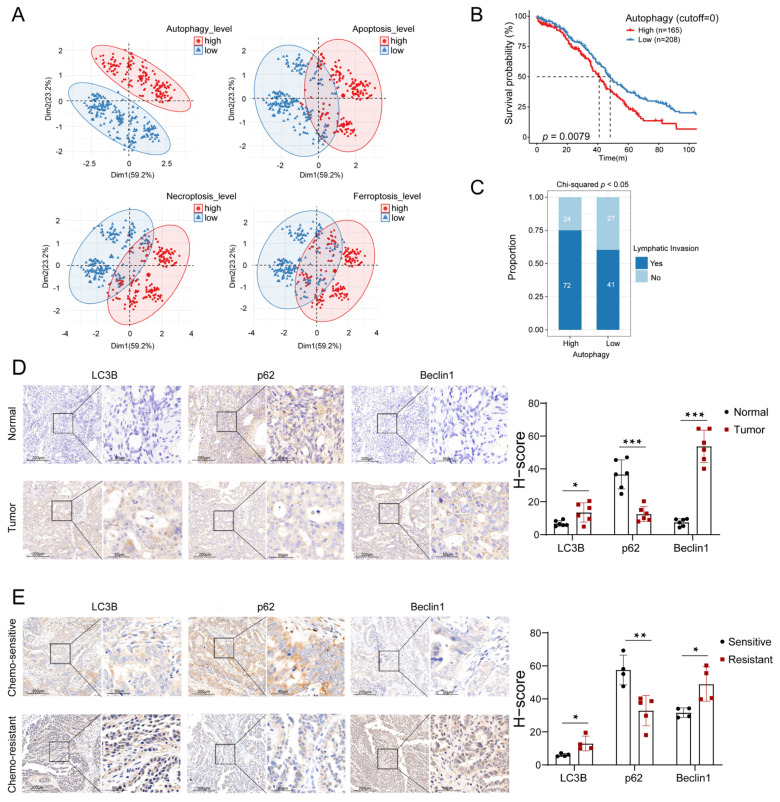
Identification and validation of the correlation between autophagy levels and prognosis of OC patients: (**A**) PCA on the correlation of the OC patients with different levels of autophagy, apoptosis, necroptosis and ferroptosis. (**B**) Kaplan-Meier analysis of the survival outcomes of patients (TCGA) with high and low autophagy levels. (**C**) The proportion of lymphatic metastases in patients with high and low autophagy levels. (**D**) Detection of the expression levels of LC3B, p62 and Beclin1 in OC and normal ovary tissues by IHC assay. (**E**) Detection of the expression levels of LC3B, p62 and Beclin1 in cisplatin-sensitive and cisplatin-resistant OC tissues by IHC assay. (Scale bar, 200 μm and 50 μm, * *p* < 0.05, ** *p* < 0.01, *** *p* < 0.001).

**Figure 2 biomolecules-13-00339-f002:**
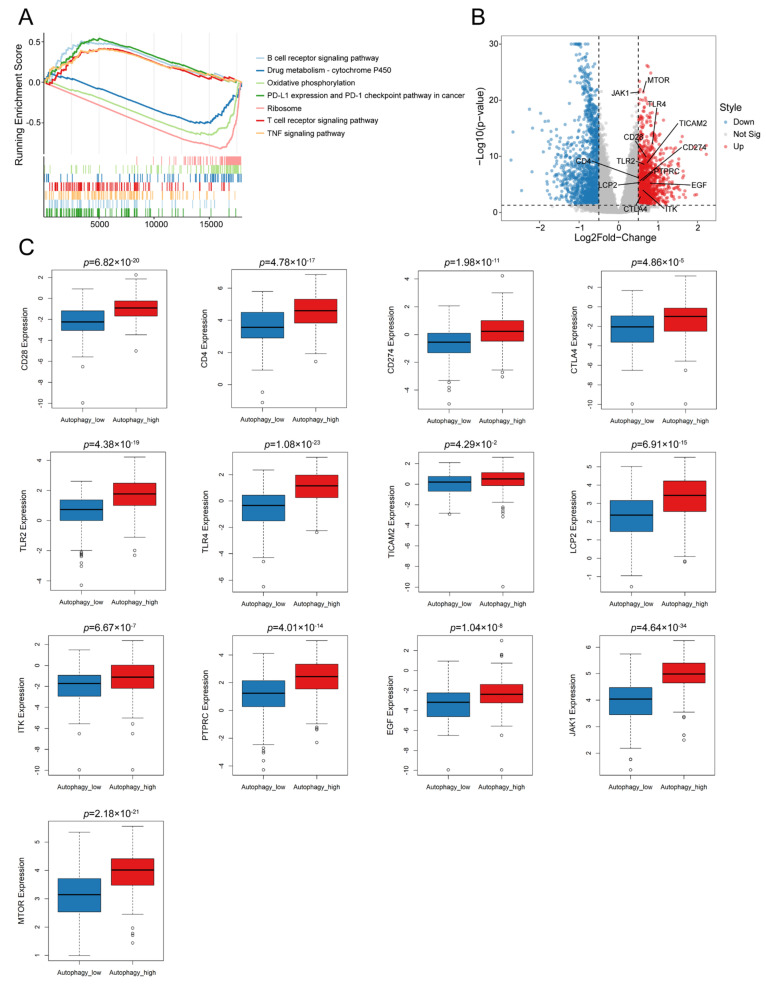
The relationship between autophagy and immune-related signaling in OC: (**A**) GSEA enrichment analysis showed immune-related signals were positively enriched in the autophagy high group. Volcano plot (**B**) and box plot (**C**) showed the expression levels of immune-related genes (CD28, CD4, CD274, CTLA4, TLR2, TLR4, TICAM2, LCP2, ITK, PTPRC, EGF, JAK1 and mTOR) in autophagy high and low group (Two-tailed student’s *t* test, *p* < 0.05 was considered to be significant).

**Figure 3 biomolecules-13-00339-f003:**
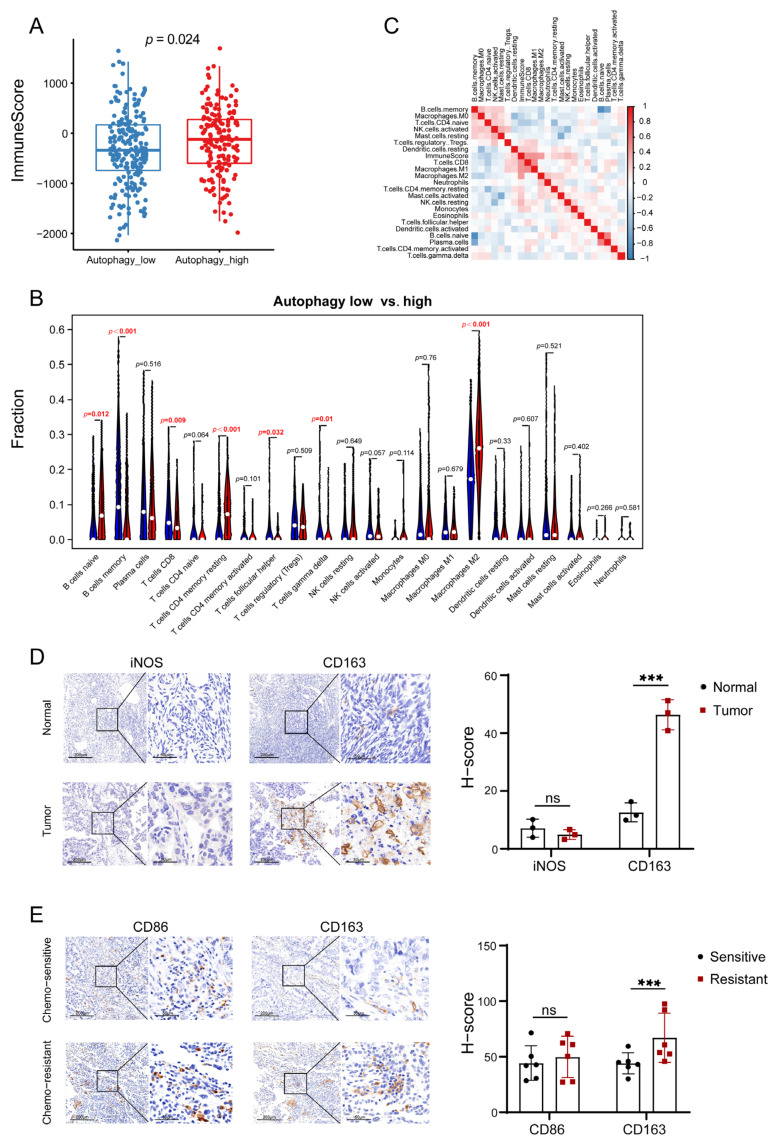
The correlation between autophagy and the levels of immune cell infiltration in OC: (**A**) Estimate analysis of the immune score in autophagy high and low group. (**B**) CIBERSORT analysis of the infiltration proportion of twenty−two immune cell types in the TIME of OC. (**C**) Correlation analysis of the differential infiltration of twenty−two immune cell types in the TIME of OC. (**D**) Detection of the expression levels of iNOS and CD163 in OC and normal ovary tissues by IHC assay. (**E**) Detection of the expression levels of CD86 and CD163 in cisplatin-sensitive and cisplatin−resistant OC tissues by IHC assay. (Scale bar, 200 μm and 50 μm; Two−tailed student’s t test, *** *p* < 0.001, ns, no statistical difference).

**Figure 4 biomolecules-13-00339-f004:**
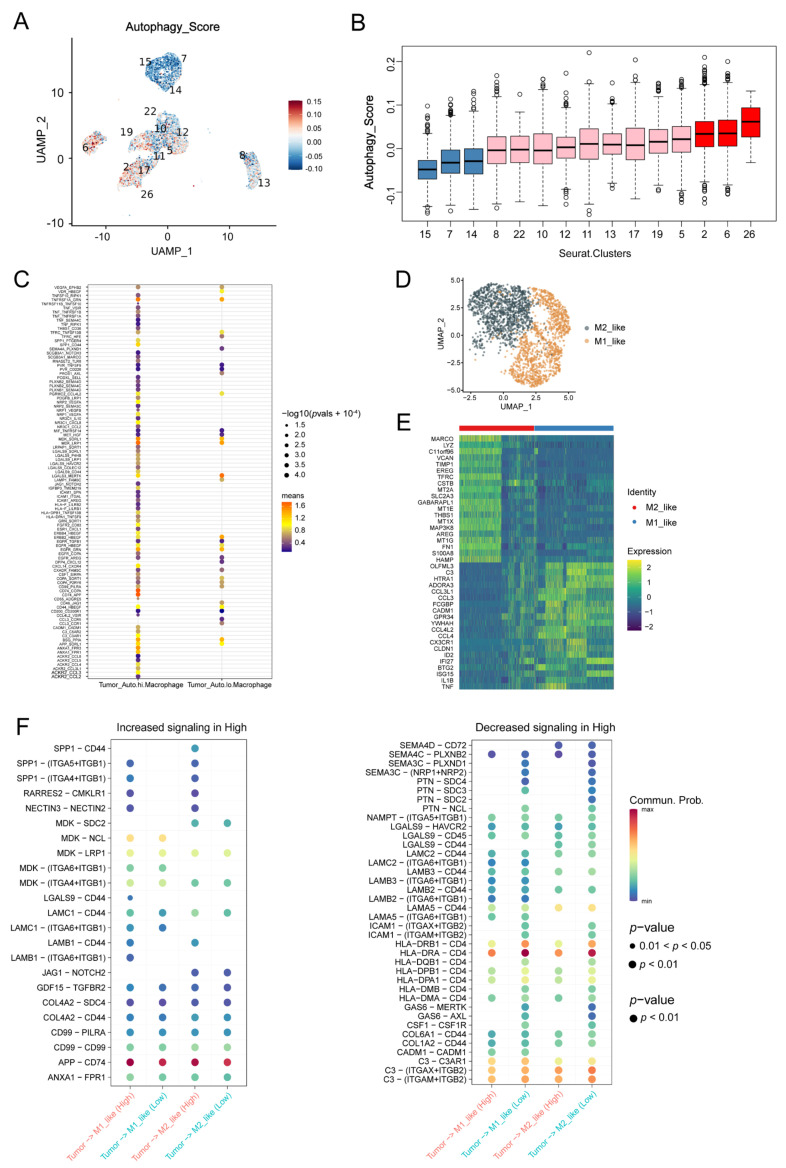
Single-cell RNA sequencing analysis of the cell-cell communication between autophagy high/low tumor cells and macrophages: UMAP plots (**A**) and Box plots (**B**) showed differential autophagy scores in different tumor cell clusters. (**C**) The ligand-receptor interactions between autophagy high/low tumor cells and macrophages. (**D**) Description of two clusters of macrophages (M1/M2) by UMAP plots. (**E**) The expression levels of the classical genes which correlated with M1/M2-like macrophages. (**F**) Different intensity of interactions between autophagy high/low tumor cells and M1/M2-like macrophages.

**Figure 5 biomolecules-13-00339-f005:**
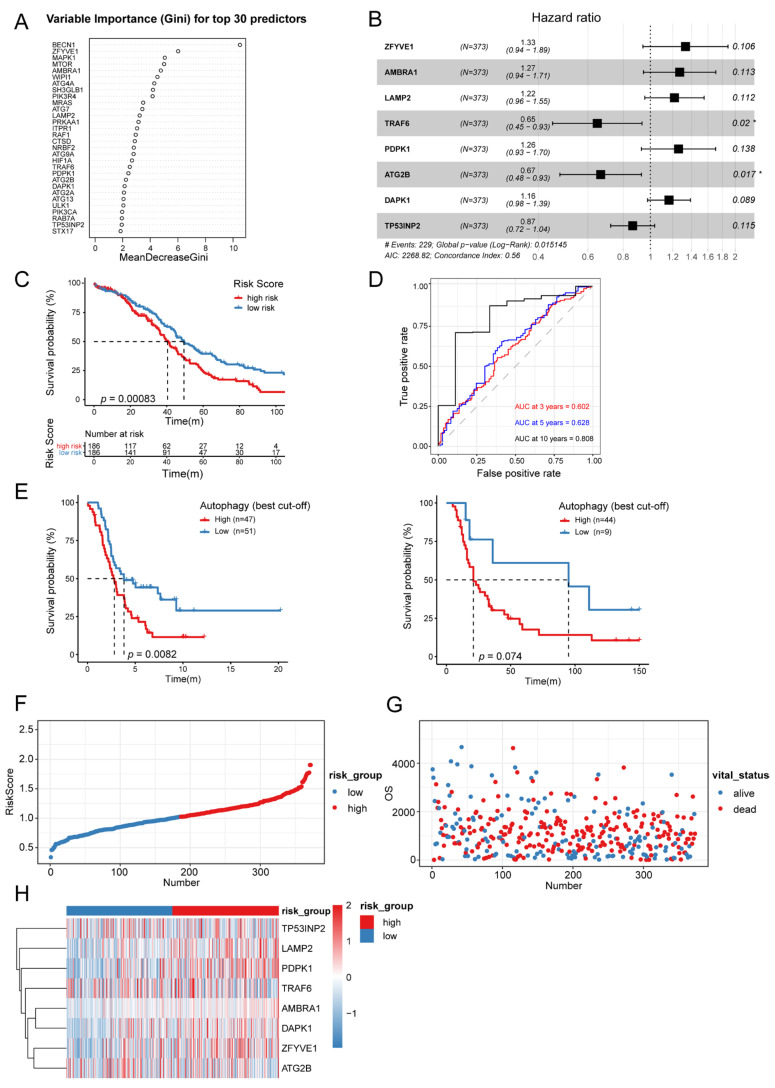
Development of an autophagy-related signature: (**A**) Ranking of top thirty autophagy hub genes based on their importance in the autophagy process. (**B**) Selection of eight risk genes by stepwise multivariate Cox proportional risk regression analysis. (**C**) Kaplan-Meier analysis of the survival outcomes of patients (TCGA) with high or low autophagy risk scores. (**D**) Time-dependent ROC curves analysis of the prognostic accuracy of the autophagy-related signature at 3 years, 5 years and 10 years. (**E**) Kaplan-Meier analysis of the survival outcomes of patients with high or low autophagy risk scores in the validation cohorts from GEO. Left: GSE14407; Right: GSE38666. (**F**) The distribution of the autophagy risk scores in OC patients from TCGA cohort. (**G**) The survival status of the OC patients in autophagy-high/low risk group. (**H**) The expression levels of the eight risk genes in autophagy-high/low risk group.

**Figure 6 biomolecules-13-00339-f006:**
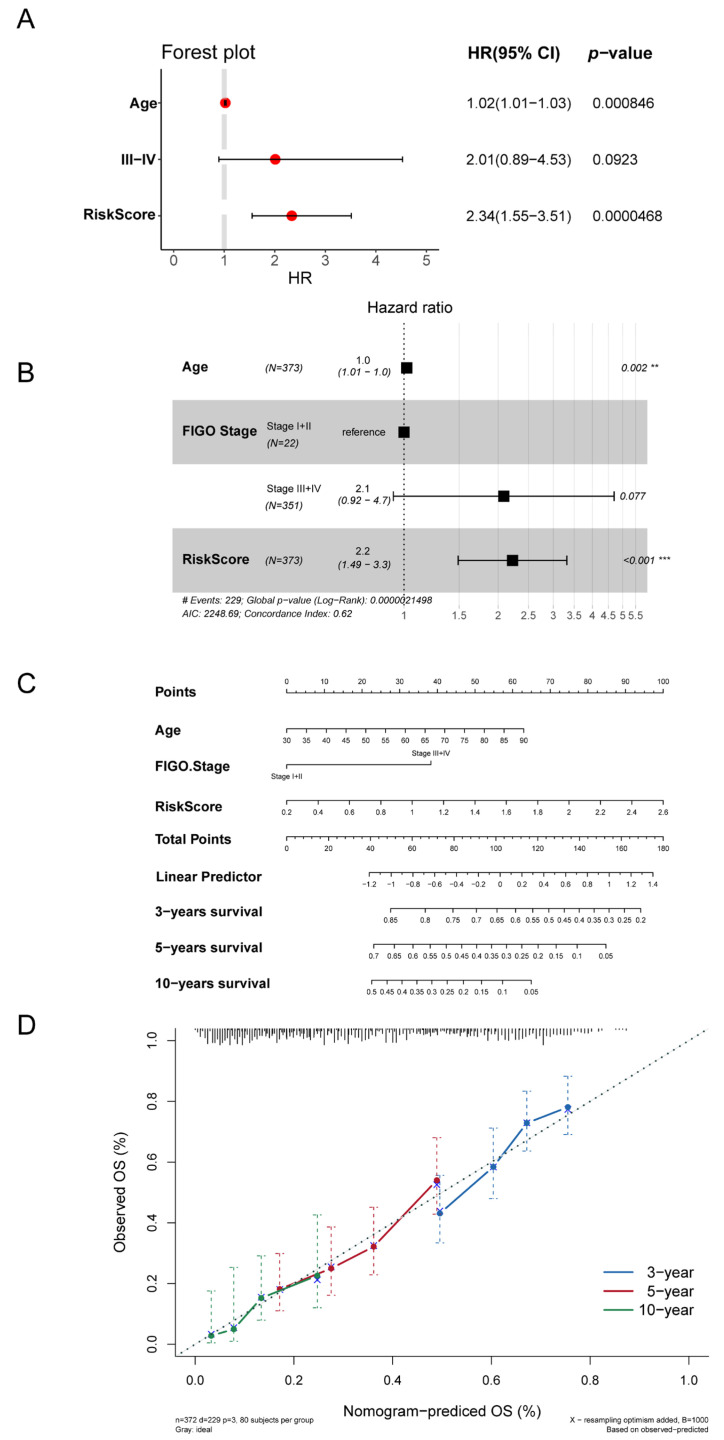
Assessment of the prognostic value of autophagy-related signature: Univariate (**A**) and multivariate (**B**) Cox regression analyses of age, FIGO stage and autophagy risk score in the TCGA cohort. (**C**) A nomogram integrating the autophagy-related signature risk score with the clinical characteristics to quantify risk evaluation for each patient. (**D**) The calibration curves for the nomogram in the OC cohorts from TCGA.

**Figure 7 biomolecules-13-00339-f007:**
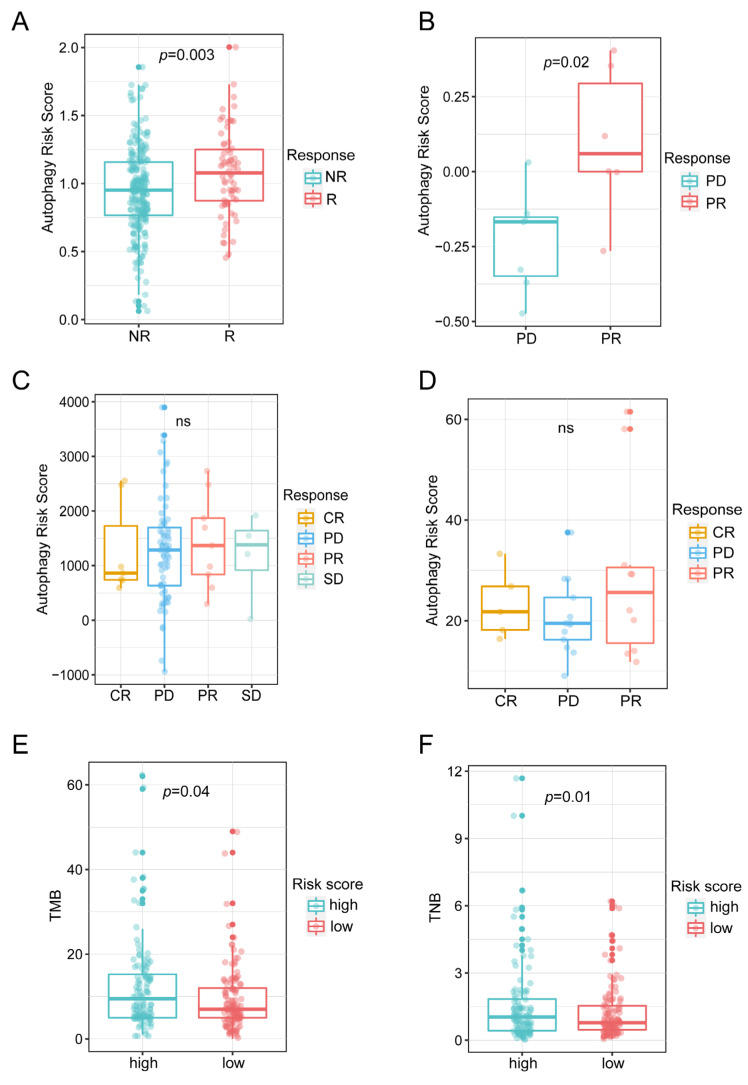
Evaluation of the ICIs therapy efficiency based on autophagy-related signature: Autophagy risk score in the patients with different responses to immunotherapy in the IMvigor210 cohort (**A**), an institutional cohort (**B**), GSE176307 (**C**) and GSE78220 cohort (**D**). TMB (**E**) and TNB (**F**) values (IMvigor210) of the patients with autophagy high/low risk score (ns, no statistical difference).

## Data Availability

All the data included in this study was from open databases except for scRNA-seq data. Further inquiries could be directed to the corresponding authors.

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
