# Peer review of "Identification of an Autophagy-Related Signature for Prognosis and Immunotherapy Response Prediction in Ovarian Cancer"

_biomolecules, 2023, doi:10.3390/biom13020339_

Round 1

Reviewer 1 Report

In this manuscript, Ding et al. used bioinformatics analysis to identify an autophagy-related signature for the prognosis and immunotherapy response prediction in ovarian cancer. As we know, ovarian cancer is one of the most malignant tumors in the female reproductive system with poor prognosis. In my opinion, this manuscript can be improved as follows,

Major:

1. A robust model need to be validated in independent cohorts; it is necessary to enroll other OV cohorts not only TCGA.

2. In Figure 1A, the author used PCA to explain that autophagy is vital to OV patients, it is a novel strategy. However, there are many programmed cell death types as we know (doi: 10.1016/j.ijsu.2022.106936). Besides, what is the cut-off value of Figure 1B.

3. Immunotherapy cohorts are other cancer types; I wonder if the author could validate the model in OV immunotherapy cohorts.

4. In Figure 4, M1/M2-like cells seem to be abandoned in many high-quality articles. Scientists prefer to use markers such as FOLR2, S100A9, TREM2, etc.

5. More bioinformatics analyses should be added.

Minor:

1. Some explanations of the abbreviation are missing, such as FPKM

2. Figure 4D, the point size seems to be too large.

3. The manuscript should be edited by a native speaker.

Reviewer 2 Report

In the manuscript “Identification of an autophagy-related signature for the prognosis and immunotherapy response prediction in ovarian cancer” the Authors aimed to identify the correlation between autophagy level and outcomes of ovarian cancer patients. They found that autophagy levels were strictly associated to the prognosis of ovarian cancer patients and to the tumor immune microenvironment status such as immune score, and immune cell infiltration.

The topic of the manuscript is interesting and up-to-date; the manuscript is written clearly and the presentation of results follows a coherent line; various methods and techniques are used.

However, I have some minor comments on the current version of the manuscript to share with the Authors.

-    - It is known that Cisplatin exerts its cytotoxic effect also by affecting ribosome biogenesis leading to nucleolar stress. In this context, several recent studies reported a correlation between autophagy and nucleolar stress in cancer cells. In this light, the authors should comment on this; it could be beneficial for the discussion. There is a related paper that might be mentioned/discussed too: Pecoraro et al. Role of Autophagy in Cancer Cell Response to Nucleolar and Endoplasmic Reticulum Stress. Int J Mol Sci. 2020 Oct 4;21(19):7334. doi: 10.3390/ijms21197334.

-   - I recommend improving the image quality. In particular, some panels are too small (e.g. Figure 5 A-D)

Reviewer 3 Report

The authors applied bioinformatics methods to clarify the role of autophagy-related genes as prognostic markers in ovarian cancer. Their observations were confirmed by IHC. Results point to a correlation between autophagy levels and the immune infiltrated cells in the tumor microenvironment. The conclusion of the work correlates a specific autophagy gene signature with poor prognosis and better immune checkpoint inhibitors response. Overall the study is well design and the proposed goals were reached. The English needs some improvement, proofreading is essential. Furthermore, some issues need to be addressed.

Major issues:

In materials 2.1. when the collection of samples is mentioned, you need to mention the project approval number from your Ethical Committee and that the study was conducted in accordance with the Declaration of Helsinki. This can also be mentioned in a separated section, you should check what are the requirements of the journal;

Mention also in 2.3. the accordance with the Declaration of Helsinki;

In 2.4. please add the antibody concentrations. Also in 2.4. detail a bit more the Image J analysis and the reference of this analysis method;

Line 179 – what do you mean by “previously self-clustered OC patients”? You publish this before? Otherwise remove previously from the phrase, please;

Line 191 – “involvement of cisplatin in prognosis” is not clear, please rephrase;

In the result section 3.1. you should include a simple explanation of what are the results of figure 1A;

In figure 1 your h score ranges from 0 to 80 but on your methods you describe the scores in a range of 1-3. Can you please explain what are the h scores in graphs D and E? Also because from the graphs, LC3B levels in sensitive patient samples seem to be in the range of normal tissue. Not sure I am looking at the graph properly, but the scale should be the same and the staining was performed in the same way, therefore are comparable? The scale bars are also not clear, and please include in the legend the bar size;

Add in figure 2. and 3. the statistical analysis performed;

In figure 3 the scale bars are also not clear, and please include in the legend the bar size;

In 3.6. include in the main text the cancer types of the other cohorts: institutional, GSE17630 and GSE78220 cohort;

Line 400 – the tumor microenvironment (TME) is composed of stromal cells such as CAFs, endothelial cells and infiltrating immune cells, not the TIME, please correct this.

Minor issues:

Line 182 – Please mention what “biological characteristics” characteristics you are looking at (like you did in fig1 legend);

In lines 226 and 228 please add the ref for these markers of TAMs M1- and M2-like;

Line 239 – were instead of was;

Line 246 – please add “classically activated M1 subtypes like subtypes”, as is not consensual that TAMs can be described in such a simplistic way. A wide range spectrum of TAMs has been described, with macrophages adopting in between status, that are not classical M1 or M2;

Line 248 – were instead of was;

Line 283 – suggestion: maybe replace “to make it possible for clinical practice” by to translate our evidence into a tool to support the clinical practice;

Line 321 – “similar histological type to OC”;

Line 332 – “benefit the patient have from ICIs therapy”;

Line 364 – suggestion: maybe replace “Usually, OC is poorly treated with surgery and chemotherapy” by usually OC is treated with surgery and chemotherapy, but with poor outcomes;

Line 399 – Instead of “Beyond” besides;

Cancer cell heterogeneity could be found, with different phenotype of cells being identified, regarding for example the autophagy levels. This is very important in the better understanding of the pathology of cancer and therapy response, and could be briefly described in the discussion;

Line 438 – “but also potentially aid in distinguishing the patient response to ICIs therapy in specific cancer types”.

Round 2

Reviewer 1 Report

No specific suggestions to authors.

Reviewer 2 Report

The authors have revised the manuscript. Thus, it is suitable for publication in the present form.

Reviewer 3 Report

My comments were addressed, thank you.